# RAIDAR: GENERATIVE AI DETECTION VIA REWRITING

**Chengzhi Mao**[1] **& Carl Vondrick**[1] **& Hao Wang**[2] **& Junfeng Yang**[1]
Columbia University[1]   Rutgers University[2]

## ABSTRACT

We find that large language models (LLMs) are more likely to modify human-written text than AI-generated text when tasked with rewriting. This tendency arises because LLMs often perceive AI-generated text as high-quality, leading to fewer modifications. We introduce a method to detect AI-generated content by prompting LLMs to rewrite text and calculating the editing distance of the output. We dubbed our gene**R**ative **AI D**etection vi**A R**ewriting method **Raidar**. Raidar significantly improves the F1 detection scores of existing AI content detection models – both academic and commercial – across various domains, including News, creative writing, student essays, code, Yelp reviews, and arXiv papers, with gains of up to 29 points. Operating solely on word symbols without high-dimensional features, our method is compatible with black box LLMs, and is inherently robust on new content. Our results illustrate the unique imprint of machine-generated text through the lens of the machines themselves.

## 1 INTRODUCTION

Large language models (LLMs) demonstrate exceptional capabilities in text generation (Cha, 2023; Brown et al., 2020; Chowdhery et al., 2022), such as question answering and executable code generation. The increasing deployment and accessibility of those LLM also pose serious risks (Bergman et al., 2022; Mirsky et al., 2022). For example, LLMs create cybersecurity threats, such as facilitating phishing attacks (Kang et al., 2023), generating propaganda (Pan et al., 2023), disseminating fake or biased content on social media, and lowering the bar for social engineering (Asfour & Murillo, 2023). In education, they can lead to academic dishonesty (Cotton et al., 2023). Pearce et al. (2022); Siddiq et al. (2022) have revealed that LLM-generated code can introduce security vulnerabilities to program. Radford et al. (2023); Shumailov et al. (2023) also find LLM-generated content is inferior to human content and can contaminate foundation models' training. Detecting and auditing those machine-generated text will thus be crucial to mitigate the potential downside of LLMs.

A plethora of works have investigated detecting machine-generated content (Sadasivan et al., 2023). Early methods, including Bakhtin et al. (2019); Fagni et al. (2021); Gehrmann et al. (2019); Ippolito et al. (2019); Jawahar et al. (2020), were effective before the emergence of sophisticated GPT models, yet the recent LLMs have made traditional heuristic-based detection methods increasingly inadequate Verma et al. (2023); Gehrmann et al. (2019). Current techniques (Mitchell et al., 2023; Verma et al., 2023) rely on LLM's numerical output metrics. Gehrmann et al. (2019); Ippolito et al. (2019); Solaiman et al. (2019) use token log probability. However, those features are not available in black box models, including state-of-the-art ones (e.g., GPT-3.5 and GPT-4). Furthermore, the high-dimensional features employed by existing methods often include redundant and spurious attributes, leading the model to overfit to incorrect features.

In this paper, we present Raidar, a simple and effective method for detecting machine-generated text by prompting LLMs to rewrite it. Similar to how humans prompt LLMs for coherent and high-quality text generation, our method uses rewriting prompts to gain additional contextual information about the input for more accurate detection.

Our key hypothesis is that text from auto-regressive generative models retains a consistent structure, which another such model will likely to also have a low loss and treat it as high quality. We observe that machine-generated text is less frequently altered upon rewriting compared to human-written text, regardless of the models used; see Figure 1 as an example. Our approach Raidar shows how

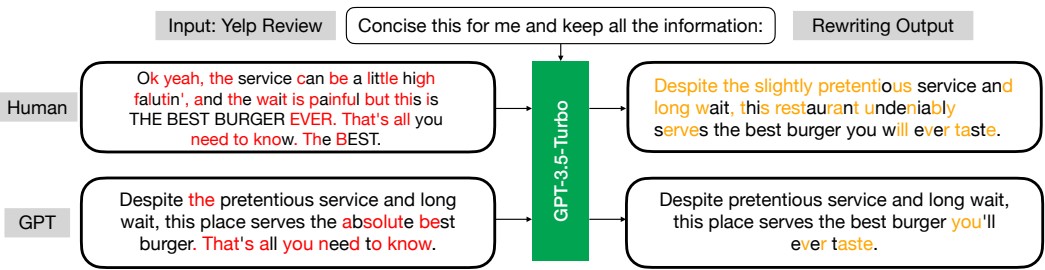

Figure 1: We introduce "Detecting via Rewriting," an approach that detects machine-generated text by calculating rewriting modifications. We show the character deletion in red and the character insertion in orange. Human-generated text tends to trigger more modifications than machine-generated text when asked to be rewritten. Our method is simple and effective, requiring the least access to LLM while being robust to novel text input.

to capitalize on this insight to create detectors for machine-generated text. Raidar operates on the symbolic word output from LLMs, eliminating the need for deep neural network features, which boosts its robustness, generalizability, and adaptability. By focusing on the character editing distance between the original and rewritten text, Raidar is semantically agnostic, reducing irrelevant and spurious correlations. This feature-agnostic design also allows for seamless integration with the latest LLM models that only provide word output via API. Importantly, our detector does not require the original generating model, allowing model A to detect the output of model B.

Visualizations, empirical experiments show that our simple rewriting-based algorithm Raidar significantly improves detection for several established paragraph-level detection benchmarks. Raidar advances the state-of-the-art detection methods (Verma et al., 2023; Mitchell et al., 2023) by up to 29 points. Our method generalizes to six different datasets and domains, and it is robust when detecting text generated from different language models, such as Ada, Text-Davinci-002, Claude, and GPT-3.5, even though the model has never been trained on text generated from those models. In addition, our detection remains robust even when the text generation is aware of our detection mechanism and uses tailored prompts to bypass our detection. Our data and code is available at https://github.com/cvlab-columbia/RaidarLLMDetect.git.

## 2 RELATED WORK

**Machine Text Generation.** Machine generated text has achieved high quality as model improves (Radford et al., 2019; Li et al., 2022; Zhou et al., 2023; Zhang et al., 2022; Gehrmann et al., 2019; Brown et al., 2020; Chowdhery et al., 2022). The release of ChatGPT enables instructional following text synthesis for the public Cha (2023). (Dou et al., 2021; Jawahar et al., 2020) demonstrate that machines can potentially leave distinctive signals in the generated text, but these signals can be difficult to detect and may require specialized techniques.

**Detecting Machine Generated Text.** Detecting AI-generated text has been studied before the emergence of LLM (Bakhtin et al., 2019; Fagni et al., 2021; Gehrmann et al., 2019; Ippolito et al., 2019). Jawahar et al. (2020) provided a detailed survey for machine-generated text detection.

The high quality of recent LLM generation makes detection to be challenging (Verma et al., 2023). Chakraborty et al. (2023) studies when it is possible to detect LLM-generated content. Tang et al. (2023) surveys literature for detecting LLM generated texts. Sadasivan et al. (2023) show that the detection AUROC is upper bounded by the gap between the machine text and human text. The state-of-the-art LLM detection algorithm (Verma et al., 2023; Mitchell et al., 2023) requires access to the probability and loss output from the LLM for the scoring model, yet those numerical metrics and features are not available for the latent GPT-3.5 and GPT-4. Mitchell et al. (2023) requires the scoring model and the target model to be the same. Ghostbuster (Verma et al., 2023) operates under the assumption that the scoring and target model are different, but it still requires access to generated documents from the target model. In addition, the output from the above deep scoring models can contain nuisances and spurious features, and can also be manipulated by adversarial attacks (Jin et al., 2019; Zou et al., 2023), making detection not robust. Another line of work aims to watermark the AI-generated text to enable detection (Kirchenbauer et al., 2023).

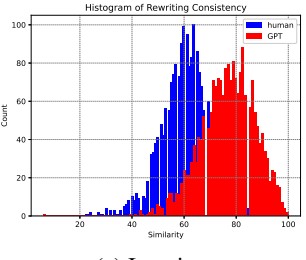 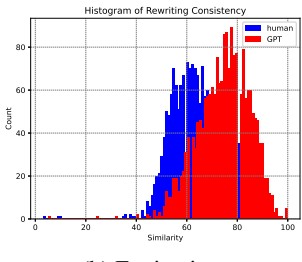 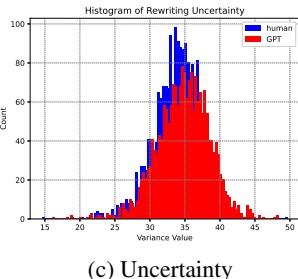

(a) Invariance          (b) Equivariance          (c) Uncertainty

Figure 2: The rewriting similarity score of human and GPT-generated text. The similarity score measures how similar the text is before and after the rewriting. A larger similarity score indicates that rewriting makes less change. (a) We show the similarity score under a single transformation; machine-generated text (red) is invariant after rewriting compared with human-generated text. (b) We show the similarity score under a transformation and its reverse transformation; the machine-generated text is more equivariant under transformation. (c) We show the uncertainty of text produced by humans and GPT. GPT input is more stable than human input. The samples are run on the Yelp Review dataset with 4000 samples. The discrepancies in invariance, equivariance, and output uncertainty allow us to detect machine-generated text.

**Bypassing Machine Text Detection.** Krishna et al. (2023) showed rephrase can remove watermark. Krishna et al. (2023); Sadasivan et al. (2023) show that paraharase can efficiently evade detection, including DetectGPT (Mitchell et al., 2023), GLTR (Gehrmann et al., 2019), OpenAI's generated text detectors, and other zero-shot methods Ippolito et al. (2019); Solaiman et al. (2019). There is a line of work that watermarks the generated text to enable future detection. However, they are shown to be easily broken by rephrasing, too. Our detection can be robust to rephrasing.

**Prompt Engineering.** Prompting is the most effective and popular strategy to adapt and instruct LLM to perform tasks Li & Liang (2021); Zhou et al. (2022); Wei et al. (2022); Kojima et al. (2022). Zero-shot GPT prompts the GPT model by asking "is the input generated by GPT" to predict if this is GPT generated (Verma et al., 2023). However, since GPTs are not trained to perform this task, they struggle. In contrast, our work constructs a few rewriting prompts to access the inherent invariance and equivariance of the input. While we can also perform an optimization-based search for better prompt (Zhou et al., 2022), we leave this for future work.

## 3   DETECTING MACHINE GENERATED TEXT BY REWRITING

We present our approach Raidar for detecting large language models generated text via rewriting. We first talk about the rewriting prompt design to access the property of the input text, then introduce our approach that detects based on the output symbolic modifications.

### 3.1   REWRITING TEXT VIA LANGUAGE MODELS AND PROMPTS

Let $F(\cdot)$ be a large language model. Given an input text $\mathbf{x}$, our goal is to classify the label $\mathbf{y}$, which indicates whether it is generated by a machine. The key observation of our method is that given the same rewriting prompt, such as asking the LLM model to "rewrite the input text," an LLM-written text will be accepted by the language model as a high-quality input with inherently lower loss, which leads to few modifications at rewriting. In contrast, a human-written text will be unfavoured by LLM and edited more by the language models.

We will use the invariance between the output and the input to measure how much LLM prefers the given input. We hypothesize that LLM will produce invariant output when rewriting its own generated text because another auto-regressive prediction will tend to produce text in a similar pattern. We define this property as the invariance property.

**Invaraince.** Given data $x$, we apply a transformation to the data via prompting the LLM with prompt $p$. If the data $x$ is produced from LLM, then the transformation $p$ that aims to rewrite the input should introduce a small change. We construct the invariance measurement as $L = D(F(p, \mathbf{x}), \mathbf{x})$, where $D(\cdot)$ denotes the modification distance.

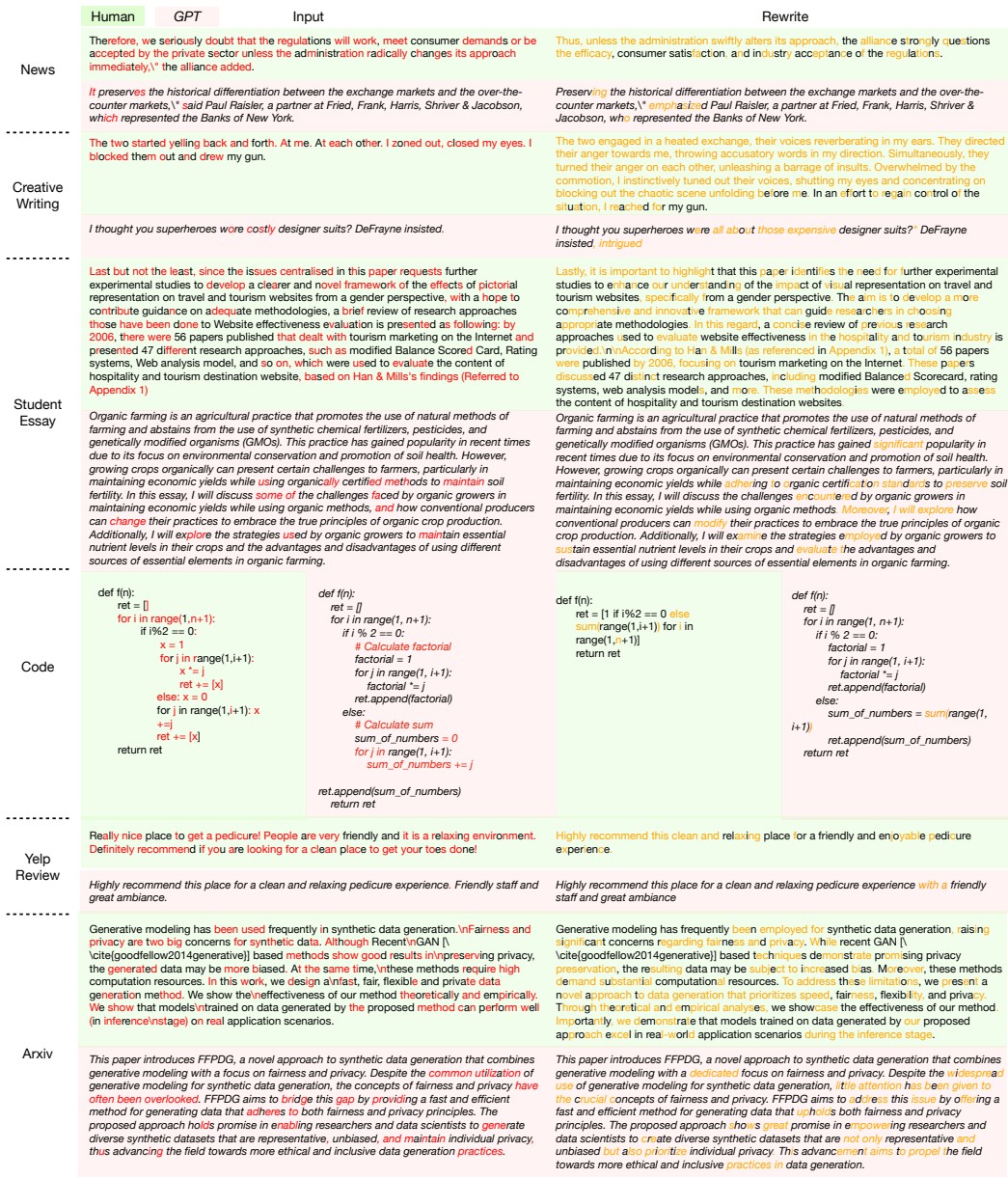

Figure 3: Examples of text rewriting on six datasets for invariance. We use a green background to indicate human-written text, and a red background to indicate machine-generated text. We show the character deletion in red and the character insertion in orange. Human-written text tends to be modified more than machine-generated text. Our detection algorithm relies on this difference to make predictions.

We manually create the prompt $p$ to access this invariance. We do not study automatic ways to generate prompts Zhou et al. (2022); Li & Liang (2021), which can be done in future work by optimizing the prompt. In this work, we will show that even a single manually written prompt can achieve a significant difference in invariance behavior. We show a few of our prompts here:

```
1. Help me polish this:
2. Rewrite this for me:
3. Refine this for me please:
```

where the goal is to make LLM modify more when rewriting human text and be more invariant when modifying LLM-generated text.

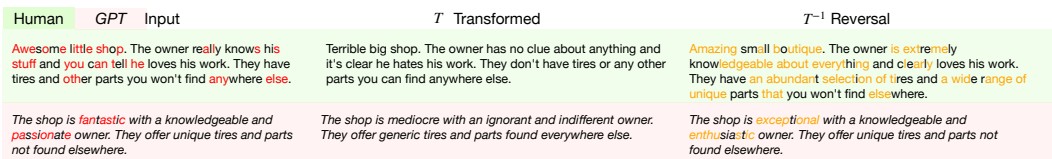

Figure 4: Examples for equivariance. We show an example on the Yelp Review dataset. For simplicity, we use identity transformation $p$, and use the "opposite meaning" as the equivariance transformation $T$. GPT data tends to be consistent to the original input after transformation and reversal.

**Equivariance.** In addition, we hypothesize that GPT data will be equivariant to the data generated by itself. Equivariance means that, if we transform the input, perform the rewriting, and undo the transformation, it will produce the same output as directly rewriting the input.

We achieve the transformation for large language models by appending a prompt $T$ to the input and asking the LLM to produce the transformed output. We denote the reversal of the transformation as $T^{-1}$, which is another prompt that writes in the opposite way as $T$. Equivariance can be measured by the following distance: $L = D(F(T^{-1}, F(p, F(T, \mathbf{x}))), F(p, \mathbf{x}))$.

Here we show two examples for the equivariance transformation prompt $T$ and $T^{-1}$:

$T$: `Write this in the opposite meaning:`
$T^{-1}$: `Write this in the opposite meaning:`

$T$: `Rewrite to Expand this:`
$T^{-1}$: `Rewrite to Concise this:`

By rewriting the sentence with the opposite meaning twice, the sentence should be converted back to its original if the LLM is equivariant to the examples. Note that this transformation $T$ is based on the language model prompt.

**Output Uncertainty Measurement.** We also assume that LLM-generated text will be more stable, when asked to rewrite multiple times than human-written text. We thus explore the variance of the output as a detection measurement. Denote the prompt to be $p$. The k-th generation results from LLM would be $\mathbf{x}'_k = F(p, \mathbf{x})$. Due to the randomness in language generation, $\mathbf{x}'_k$ will be different. We denote the editing distance between two outputs $A$ and $B$ as $D(A, B)$. We construct the uncertainty measurement as: $U = \sum_{i=1}^{K-1} \sum_{j=i}^{K} D(\mathbf{x}'_i, \mathbf{x}'_j)$. Note that, in contrast to the invariance and equivariance, this metric only uses the output, and the original input is not in the calculation of the output uncertainty.

### 3.2 Measuring Change in Rewriting

We treat the output of LLM as symbolic representations that encode information about the data. In contrast to Mitchell et al. (2023); Verma et al. (2023), our detection algorithm does not use continuous, numerical representations of the word tokens. Instead, our algorithm operates totally on the discrete, symbolic representations from the LLM. By prompting LLM, our method obtains additional information about the input text via the rewriting difference. We will show how to measure the rewriting change below:

**Bag-of-words edit.** We use the change of bag-of-words to capture the edit created by LLM. We compute the number of common bags of n-words divided by the length of the input.

**Levenshtein Score**. Levenshtein score (Levenshtein, 1966) is a popular metric for measuring the minimum number of single-character edits, including deletion and addition, to change one string to the other. We use standard dynamic programming to calculate the Levenshtein distance. A higher score denotes the two strings are more similar. We use Levenshtein$(A, B)$ to denote the edit distance between string $A$ and $B$. Let the rewriting output $\mathbf{s}_k = F(p_k, \mathbf{x})$. We obtain the ratio via:

$$D_k(\mathbf{x}, \mathbf{s}_k) = 1 - \frac{\text{Levenshtein}(\mathbf{s}_k, \mathbf{x})}{\max(len(\mathbf{s}_k), len(\mathbf{x}))}.$$

We use ratio because the feature of editing difference should be independent of the text length. The invariance, equivariance, and uncertainty measured by the above metric will be used as features for

Table 1: F1 score for detecting machine-generated paragraphs. The results are in domain testing, where the model has been trained on the same domain. We **bold** the best performance on in-distribution and out-of-distribution detection. Our method achieved over 8 points of improvement over the established state-of-the-art.

| Methods | News | Datasets Creative Writing | Student Essay | Code | Yelp Reviews | Arxiv Abstract |
|---|---|---|---|---|---|---|
| GPT Zero-Shot Verma et al. (2023) | 54.74 | 20.00 | 52.29 | 62.28 | 66.34 | 65.94 |
| GPTZero (Tian, 2023) | 49.65 | 61.81 | 36.70 | 31.57 | 25.00 | 45.16 |
| DetectGPT Mitchell et al. (2023) | 37.74 | 59.44 | 45.63 | 67.39 | 69.23 | 66.67 |
| Ghostbuster Verma et al. (2023) | 52.01 | 41.13 | 42.44 | 65.97 | 71.47 | 76.82 |
| Ours (Invariance) | **60.29** | **62.88** | **64.81** | **95.38** | **87.75** | 81.94 |
| Ours (Equivariance) | 58.00 | 60.27 | 60.07 | 80.55 | 83.50 | 75.74 |
| Ours (Uncertainty) | 60.27 | 60.27 | 57.69 | 77.14 | 81.79 | **83.33** |

Table 2: F1 score for detecting machine-generated paragraph following the out-of-distribution setting in Verma et al. (2023). We use logistic regression classifier for all ours. Our method achieved over 22 points of improvement over the established state-of-the-art.

| Methods | News | Datasets Creative Writing | Student Essay |
|---|---|---|---|
| Ghostbuster Verma et al. (2023) | 34.01 | 49.53 | 51.21 |
| Ours (Invariance) | 56.47 | 55.51 | **52.77** |
| Ours (Equivariance) | **56.87** | **59.47** | 51.34 |
| Ours (Uncertainty) | 55.04 | 52.01 | 47.47 |

a binary classifier, which predicts the generation source of the text. For details of the algorithm, please refer to Appendix A.3.

Our design enjoys several advantages. First, since we only access the discrete token output from LLM, our algorithm requires minimal access to the LLM models. Given that the major state-of-the-art LLM models, like GPT-3.5-turbo and GPT-4 from OpenAI, are black-box models and only provide API for accessing the discrete tokens rather than the probabilistic values, our algorithm is general and compatible with them. Second, since our representation is discrete, it is more robust in the sense that it will be invariant to the perturbations and shifting in the input space. Lastly, our symbolic representations enable us to construct the following measurements that are none differentiable, which introduces extra burden and cost for gradient-based adversarial attempts to bypass our detection model.

## 4 RESULTS

We conduct experiments on detecting AI-generated text on paragraph level and compare it to the state of the art. To further understand factors that affect detection performance, we also study the robustness of our method under input aiming to evade our detection, detection accuracy on text generated from different LLM sources, and evaluate our method with different LLM for rewriting.

### 4.1 DATASET

To evaluate our approach to the challenging, paragraph-level machine-generated text detection, we experiment with the following datasets.

**Creative Writing Dataset** is a language dataset based on the subreddit WritingPrompts, which is creative writing by a community based on the prompts. We use the dataset generated by Verma et al. (2023). We focus on detecting paragraph-level data, which is generated by text-davinci-003.

**News Dataset** is based on the Reuters 50-50 authorship identification dataset. We use the machine-generated text from Verma et al. (2023) via text-davinci-003.

Table 3: Performance under adaptive prompts aiming to evade our detector. In the "Single Training Prompt" column, the detector is trained on a non-adaptive prompt and tested against both the same prompt and two evasive prompts. Adversarial rephrasing can bypass our detector. In "Multi Training Prompt*", the model is trained using two prompts and tested on a third, different prompt. The last two rows shows results under adaptive prompts to evade our detection. Training on multiple prompts enhances our detector's robustness against machine-generated inputs attempting evasion.

| Test Prompt | Single Training Prompt | | | Multi Training Prompt* | | |
|---|---|---|---|---|---|---|
| | Code | Yelp | Arxiv | Code | Yelp | Arxiv |
| No Adaptive Prompt | 95.38 | 87.75 | 81.94 | 92.76 | 58.04 | 82.25 |
| Prompt 1 to bypass detection | 34.15 | 61.38 | 43.81 | 86.95 | 69.19 | 91.89 |
| Prompt 2 to bypass detection | 25.64 | 61.38 | 50.90 | 88.88 | 73.23 | 93.06 |

**Student Essay Dataset** The dataset is based on the British Academic Written English corpus and generated by Verma et al. (2023).

**Code Dataset.** The goal is to detect if the Python code has been written by GPT, which can be important for education. We adopt the HumanEval dataset (Chen et al., 2021) as the human-written code, and ask GPT-3.5-turbo to perform the same task and generate the code.

**Yelp Review Dataset.** Yelp reviews tend to be short and challenging to detect. We use the first 2000 human reviews from the Yelp Review Dataset, and generate concise reviews via GPT-3.5-turbo in a similar length as the human written one.

**ArXiv Paper Abstract.** We investigate if we can detect GPT written paragraphs in academic papers. Our dataset contains 350 abstracts from ICLR papers from 2015 to 2021, which are human-written texts since ChatGPT was not released then. We use GPT-3.5-turbo to generate an abstract based on the paper's title and the first 15 words from the abstract.

## 4.2 BASELINES

**GPT Zero-shot** (Verma et al., 2023) performs detection by directly asking GPT if the input is written by GPT or not. We use the same prompt as Verma et al. (2023) to query GPT.

**GPTZero** (Tian, 2023) is an commercial machine text detection service.

**DetectGPT** (Mitchell et al., 2023) is the state-of-the-art thresholding approach to detect GPT-generated text, which achieved 99-point performance over a longer input context, yet its performance on shorter text is unknown. It thresholds the curvature of the input to perform detection. We use the facebook/opt-2.7B for the scoring model.

**Ghostbuster** (Verma et al., 2023) is the state-of-the-art classifier for machine generated text detection. It uses probabilistic output from large language models as features, and performs feature selection to train an optimal classifier.

## 4.3 MAIN RESULTS

We use GPT-3.5-Turbo as the LLM to rewrite the input text. Once we obtain the editing distance feature from the rewriting, we use Logistic Regression (Berkson, 1944) or XGBoost (Chen & Guestrin, 2016) to perform the binary classification. We compare our results on three datasets from Verma et al. (2023), as well as our created three datasets, in Table 15. Our method Raidar outperforms the Ghostbuster method by up to 29 points, which achieves the best results over all baselines. In Table 2, we follow the out-of-distribution (OOD) experiment setup in Verma et al. (2023), where we trained the detection classifier on one dataset and evaluated on the other. For the OOD experiment, our method still improves by up to 32 points, demonstrating the effectiveness of our approach over prior methods.

Table 4: Robustness in detecting outputs from various language models. Using the same GPT-3.5-Turbo rewriting model, we present F1 detection scores for detecting text from five generation models across three diverse tasks. In the in-distribution experiment, detectors are trained and tested on the same model. For out-of-distribution, detectors are trained on text from other generators. Overall, our method effectively detects machine-generated text in both scenarios.

| LLM Model Used for Text Generation | Raidar (Ours) | | | | | | DetectGPT | | |
|---|---|---|---|---|---|---|---|---|---|
| | In Distribution | | | Out of Distribution | | | | | |
| | Code | Yelp | arXiv | Code | Yelp | arXiv | Code | Yelp | arXiv |
| Ada | 96.88 | **96.15** | 97.10 | 62.06 | 72.72 | 70.00 | 67.39 | 70.59 | 69.74 |
| Text-Davinci-002 | 84.85 | 65.80 | 76.51 | 75.41 | 51.06 | 60.00 | 66.82 | 71.36 | 66.67 |
| GPT-3.5-turbo | 95.38 | 87.75 | 81.94 | **91.43** | 71.42 | 48.74 | 67.39 | 69.23 | 66.67 |
| GPT-4-turbo | 80.00 | 83.42 | 84.21 | 83.07 | 79.73 | 74.02 | 70.97 | 66.94 | 66.99 |
| LLaMA 2 | **98.46** | 89.31 | **97.87** | 70.96 | **89.30** | **74.41** | 68.42 | 67.24 | 66.67 |

Table 5: Effectiveness of detection using various large language models for rewriting. We present detection F1 scores for the same input data rewritten by Ada, Text-Davinci-002, and GPT-3.5. Among these, GPT-3.5-turbo yields the highest performance in rewriting for detection.

| LLM for Rewriting | Datasets | | | | | |
|---|---|---|---|---|---|---|
| | News | Creative Writing | Student Essay | Code | Yelp | Arxiv |
| Ada | 55.73 | 62.50 | 57.02 | 77.42 | 73.33 | 71.75 |
| Text-Davinci-002 | 55.47 | 60.59 | 58.96 | 82.19 | 75.15 | 59.25 |
| GPT 3.5 turbo | **60.29** | **62.88** | **64.81** | **95.38** | **87.75** | **81.94** |
| LLaMA 2 | 56.26 | 61.88 | 60.48 | 85.33 | 74.85 | 72.59 |

## 4.4 ANALYSIS

**Detection Robustness against Rephrased Text Generation to Evade Detection.** Krishna et al. (2023); Sadasivan et al. (2023) show that paraphrasing can often evade detection. In Table 15, we show that our approach can detect GPT text when they are not adversarially rephrased. However, a sophisticated adversary might craft prompts for GPT such that the resulting text, when rewritten, undergoes significant changes, thereby evading our detection. We modify the GPT input using the following rephrases:

```
1. Help me rephrase it in human style
2. Help me rephrase it, so that another GPT rewriting will cause a lot
   of modifications
```

Table 3 reveals that while our detector, trained on the default single prompt data, can be bypassed by adversarial rephrasing (left columns). In the right columns, we show results when trained on two of the prompts and tested on the remaining prompts. The detectors are trained on multi-prompt data, which enhances its robustness. Even when tested against unseen adversarial prompts, our detector still identifies machine-generated content designed to elude it, achieving up to 93 points on F1 score. One exception is on the Yelp dataset; the "no adaptive prompt" has lower performance on "multiple training prompts" than "single training prompts". We suspect it is due to the Yelp dataset introducing a larger data difference when prompted differently, and this "multiple training prompts" setup will decrease performance due to training and testing on different prompts. In general, results in Table 3 demonstrate that with proper training, our method can be still robust under rephrased text to evade detection, underscoring the significance of diversifying prompt types when learning our detector.

**Source of Generated Data.** In our main experiment, we train our detector on text generated from GPT-3.5. We study if our model can still detect machine-generated text when they are generated from a different language model. In Table 4, we conduct experiments on text generated from Ada, text-davinci-002, and GPT-3.5 model. For all experiments, we use the same GPT-3.5 to rewrite.

For in-distribution experiments, we train the detector on data generated from the respective language model. Despite all rewrites being from GPT-3.5, we achieved up to 96 F1 score points. Notably, GPT-3.5 excels at detecting Ada-generated content, indicating our method's versatility in identifying

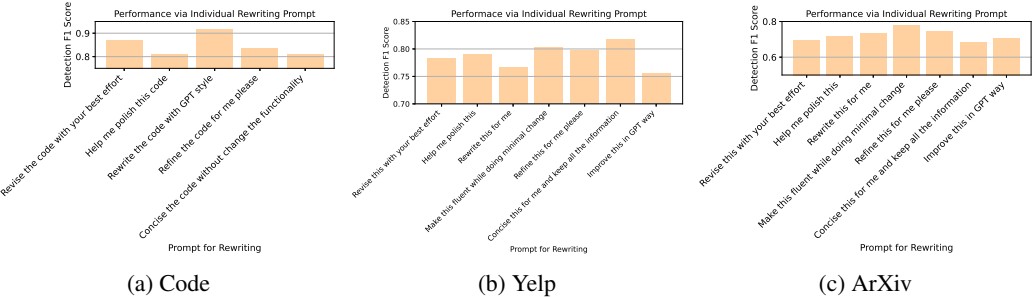

|  |  |  |
|---|---|---|
| (a) Code | (b) Yelp | (c) ArXiv |

Figure 6: Performance of individual prompt. Different prompts used during rewriting can have a significant impact on the final detection performance. There is no single prompt that performs best across all data sources. With a single rewriting prompt, we can obtain up to 90 points of detection F1 score.

both low (Ada) and high-quality (GPT-3.5) data, even they are generated from a different model. We also evaluate our detection efficiency on the Claude (Anthropic, 2023) generated text on student essay (Verma et al., 2023), where we achieve an F1 score of 57.80.

In the out-of-distribution experiment, we train the detector on data from two language models, assuming it is unaware that the test text will be generated from the third model. Despite a performance drop on detecting the out-of-distribution test data generated from the third model, our method remains effective in detecting content from this unseen model, underscoring our approach's robustness and adaptability, with up to 91 points on F1 score.

**Type of Detection Model.** Mireshghallah et al. (2023) showed that model size affects performance in perturbation-based detection methods. Given the same input text generated from GPT-3.5, We explore our approach's efficacy with alternative rewriting models with different size. In addition to using the costly GPT-3.5 to rewrite, we incorporate two smaller models, Ada and Text-Davinci-002, and evaluate their detection performance when they are used to rewrite. In Table 5, while all models achieve significant detection performance, our results indicate that a larger rewriting language model enhances detection performance in our method.

**Impact of Different Prompts.** Figure 6 displays the detection F1 score for various prompts across three datasets. While Mitchell et al. (2023) employs up to 100 perturbations to query LLM and compute curvature from loss, our approach achieves high detection performance using just a single rewriting prompt.

**Impact of Content Length.** We assess our detection method's performance across varying input lengths using

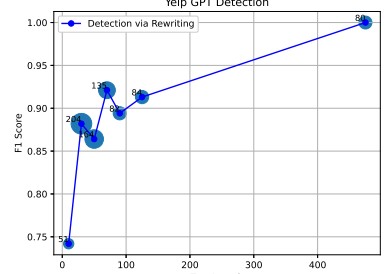

Figure 5: Detection performance as input length increases. On the Yelp dataset, we show that longer input often enables better detection performance. The number shows the number of data, reflecting by the size of the dot.

the Yelp Review dataset in Figure 5. Longer inputs, in general, achieve higher detection performance. Notably, while many algorithms fail with shorter inputs (Tian, 2023; Verma et al., 2023), our method can achieve 74 points of detection F1 score even with inputs as brief as ten words, highlighting the effectiveness of our approach.

## 5 CONCLUSION

We introduce Raidar, an approach to use rewriting editing distance to detect machine-generated text. Our results demonstrate improved detection performance across several benchmarks and state-of-the-art detection methods. Our method is still effective when detecting text generated from novel language models and text generated via prompts that aim to bypass our detection. Our findings show that integrating the inherent structure of large language models can provide useful information to detect text generated from those language models, opening up a new direction for detecting machine-generated text.

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

# A APPENDIX

## A.1 DATA CREATION

Our dataset selection was driven by the need to address emerging challenges and gaps in current research. We incorporated news, creative writing, and essays from the established Ghostbuster Verma et al. (2023) to maintain continuity with prior work. Recognizing the growing capabilities of Language Learning Models (LLMs) like ChatGPT in generating code, and the accompanying security issues, we included code as a novel and pertinent text data type. Additionally, we analyzed Yelp reviews to explore LLMs' potential for generating fake reviews, a concern overlooked in previous studies, which could significantly influence public opinion about businesses. Lastly, we included arXiv data to address recent concerns over the use of GPT in academic writing, reflecting on its ethical implications.

**Code Dataset.** Human eval dataset offers code specification and the completed code for each data point. We first use GPT to generate a detailed description of the function of the code by prompting it with "Describe what this code does code specificationcode". The result, termed pseudo code, is an interpretation of the code. Subsequently, we prompt GPT with "I want to do this pseudo code, help me write code starting with this code specification," to generate Python code that adheres to the given input-output format and specifications. This way, we create the AI-generated code data.

**Yelp Reviews Dataset.** When tasked with crafting a synthetic Yelp review, prompting GPT-3.5 with "Help me write a review based on this original review" resulted in verbose and lengthy text. However, we discovered that using the prompt "Write a very short and concise review based on this: original review" yielded the most effective and succinct AI-generated reviews.

**ArXiv Dataset.** In our experiment with Arxiv data, which includes titles and abstracts, we synthesized abstracts by using the title and the first 15 words of the original abstract. We employed the prompt "The title is title, start with first 15 words, write a short concise abstract based on this:", which successfully generated realistic abstracts that align with the titles.

## A.2 DATASET STATISTICS

In Table 6 and Table 7, we show each dataset's size, median, min, and max length on human-written and machine-generated ones, respectively.

Table 6: Statistics for each dataset from humans. We show the length in word count. Our work focuses on detecting paragraph-level text, which generally has a shorter and more challenging length.

| | News | Creative Writing | Datasets Student Essay | Code | Yelp | Arxiv |
|---|---|---|---|---|---|---|
| Dataset Size | 730 | 973 | 22172 | 164 | 2000 | 350 |
| Median Length | 38 | 21 | 96 | 96 | 21 | 102 |
| Minimum Length | 2 | 2 | 16 | 2 | 6 | 19 |
| Maximum Length | 122 | 295 | 1186 | 78 | 1006 | 274 |

## A.3 ALGORITHM

We show the algorithm for invariance, equivariance, and uncertainty based algorithms. We denote the learned classifier as $C$.

## A.4 ANALYSIS

**Quality of the Machine-Generated Content.** LLM tends to treat the text generated by the machine as high quality and conducts few edits. We conduct a human study on whether the text generated by machines is indeed of higher quality than that written by humans. This study focused on the Yelp and Arxiv datasets. Participants were presented with two pieces of text designed for the same

Table 7: Statistics for each dataset generated by GPT-3.5-Turbo. We show the length in word count. Our work focuses on detecting paragraph-level text, which generally has a shorter and more challenging length.

| | News | Creative Writing | Datasets Student Essay | Code | Yelp | Arxiv |
|---|---|---|---|---|---|---|
| Dataset Size | 479 | 728 | 13629 | 164 | 2000 | 350 |
| Median Length | 45 | 38 | 82 | 35 | 48 | 72 |
| Minimum Length | 3 | 2 | 2 | 5 | 2 | 15 |
| Maximum Length | 208 | 354 | 291 | 182 | 227 | 129 |

---

**Algorithm 1** Detecting LLM Generated Content via Output Invariance

---

1: **Input:** Text input $\mathbf{x}$, rephrase prompt $\mathbf{P}_k$, where $k = 1, ..., K$.
2: **Output:** Class prediction $\hat{y}$
3: **Inference:**
4: **for** $k = 1, ..., K$ **do**
5:     Obtain LLM output $S_k = F(\mathbf{P}_k, \mathbf{x})$
6:     Calculate bag-of-words edit $R_k$ and the Levenshtein Score $D_k$
7: **end for**
8: Make final prediction via $y = C([R_1, R_2, ..., R_K, D_1, D_2, ..., D_K])$

---

purpose, one authored by a human and the other by a machine, and were asked to judge which was of higher quality. The study involved three users, and for each dataset, we randomly selected 20 examples for evaluation. The results, detailed in Table 8, generally indicate that human-written texts are of similar or higher quality compared to those generated by machines.

Table 8: Human study on the quality of machine generated text. Our work showed that machine generated text will be perferred by LLMs and produce few edits when asked to rewrite. We also evaluate the ratio of machine generated text that is perferred by human. The machine is good at creating realistic Yelp reviews, but not good at academia paper writing.

| Methods | Yelp Reviews | Arxiv Abstract |
|---|---|---|
| % that Machine Generated Text are Preferred Human Written Text | 53.3% | 26.7% |

**Robustness of Our Method to LLM Fine-tuning.** We run the experiment on GPT-3.5-Turbo and GPT-4-Turbo. GPT-4-Turbo can be roughly treated as a realistic, advanced, continual fine-tuned LLM on new real-world data from GPT-3.5-Turbo. We show the results in Table 9. Our method is robust to LLM finetuned. Despite a drop in detection performance, it still outperforms the established state-of-the-art zero-shot detector.

**Robustness of Our Method to Non-native Speaker.** Prior work showed that LLM detectors are biased against non-native English writers, because non-native English writing is limited in linguistic expressions and is often detected as AI-generated Liang et al. (2023). We investigate if our approach can detect non-native English writers better or if it is biased against them, as shown by prior detection methods.

Following the setup from Liang et al Liang et al. (2023), we use the Hewlett Foundation's Automated Student Assessment Prize (ASAP) dataset and adopt the first 200 datasets in our study, which is a dataset from non-native speakers on TOEFL essays on 8-th grade level in the US. We create the machine-generated answer for the TOEFL essay via the following prompt:

```
Write an essay based on this:
```

---

**Algorithm 2** Detecting LLM Generated Content via Output Equivariance

---

1: **Input:** Text input $\mathbf{x}$.
2: **Output:** Class prediction $\hat{y}$
3: **Inference:**
4: **for** $k = 1, ..., K$ **do**
5:     Create transformation prompt $\mathbf{T}_k$ and inverse transformation prompt $\mathbf{T}'_k$, create rephrase prompt $P_k$.
6:     Obtain LLM output $\mathbf{M}_k = F(\mathbf{T}_k, \mathbf{x})$
7:     Obtain LLM output $\mathbf{M}'_k = F(\mathbf{P}_k, \mathbf{M}_k)$
8:     Obtain LLM output $S_k = F(\mathbf{T}'_k, \mathbf{M}'_k)$
9:     Calculate bag-of-words edit $R_k$ and the Levenshtein Score $D_k$
10: **end for**
11: Make final prediction via $y = C([R_1, R_2, ..., R_K, D_1, D_2, ..., D_K])$

---

**Algorithm 3** Detecting LLM generated Content via Output Uncertainty

---

1: **Input:** Text input $\mathbf{x}$.
2: **Output:** Class prediction $\hat{y}$
3: **Inference:**
4: Given rephrase prompt $\mathbf{P}$
5: **for** $k = 1, ..., K$ **do**
6:     Obtain LLM output $S_k = F(\mathbf{P}, \mathbf{x})$
7: **end for**
8: **for** $k = 1, ..., K$ **do**
9:     **for** $j = k, ..., K$ **do**
10:         Calculate bag-of-words edit $R_{k,j}$ and the Levenshtein Score $D_{k,j}$
11:     **end for**
12: **end for**
13: Make final prediction via $y = C([R_{1,2}, R_{1,3}, ..., R_{K-1,K}, D_{1,2}, D_{1,3}, ..., R_{K-1,K}])$

---

We show the detection result in Table 10. Our method does not discriminate the non-English speaker, and reaches a similar level of detection performance on high-quality writing (abstract from accepted ICLR papers). Since both ASAP and Arxiv are written by humans, they will be treated as low-quality text that does not match the inherent inertia in LLM models, and thus will both be modified more than the machine-generated text. Our detection algorithm will classify those texts with more modifications than humans. Thus, both non-native and efficient writers will be correctly classified by our approach. Since our algorithm only relies on the word edit distance, it does not rely on the superficial semantics of the text for detection. Thus, our approach generalizes well from academic ICLR abstract to non-native English writing on the 8th grade level, with only less than 1 point of performance drop.

**Detection Performance by combining rewrites from multiple LLMs.** In Table 11, we show detection performance when combining GPT-3.5 rewrites with other LLMs, including Ada, Davinci, and both. We find combining rewriting from multiple LLMs can improve performance over Arxiv detection, but not on Yelp.

**Detection performance by adding edit distance between the rewritten texts from different LLMs as additional features.** In Table 12, we show the detection performance. We can achieve better detection performance leveraging this new feature.

**Detection performance by combining features of invariance, equivariance, and uncertainty.** We conduct experiments in Table 13, on the two dataset we studied, we cannot further improve performance.

**Detection performance under different input length.** We show the trend in Figure 7, Figure 8, and Figure 9.

**Statistical significance of the number of changes (deletions, insertions) done by the selective generative models between humans and machine-generated texts.** We calculate the t-statistic

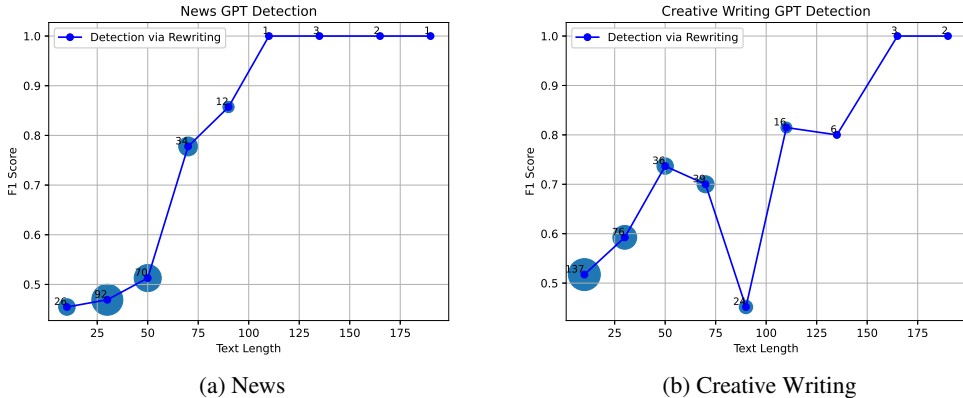

(a) News

(b) Creative Writing

Figure 7: Detection performance under different length. For News and Creative Writing datasets, longer length helps detection.

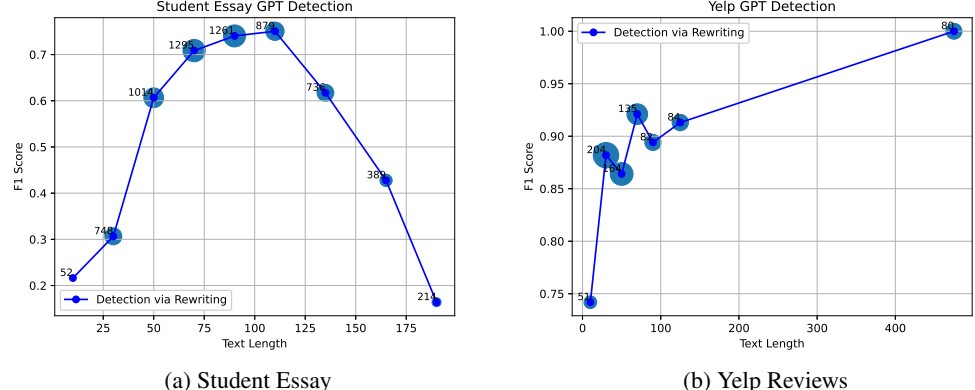

(a) Student Essay

(b) Yelp Reviews

Figure 8: Detection performance under different length. For both datasets, longer length helps detection. Yet, on student essay, input longer than 125 words will lead to performance degradation.

and calculate the p-value. In Table 14, we show the p-value for the two distributions shown in Figure 2. Since the p-value is much smaller than 0.05, it demonstrates that the number of changes between human and machine-generated text is significant.

## A.5 IMPLEMENTATION DETAILS

**The training and testing domain for Table 2.** For all experiments in Table 2, we use logistic regression, and use the same source and target for invariance, equivariance, and uncertainty. For News, we train on Creative Writing and test on News. For Creative Writing, we train on News and test on Creative Writing. FOr Student Essay, we train on News, and test on student Essay.

**Classifier choice for Table 1 and Table 2.** We use logistic regression for all our experiments except for on the student essay dataset, where we find XGBoost achieves better performance.

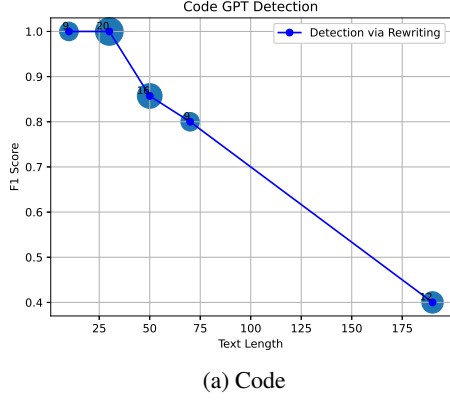
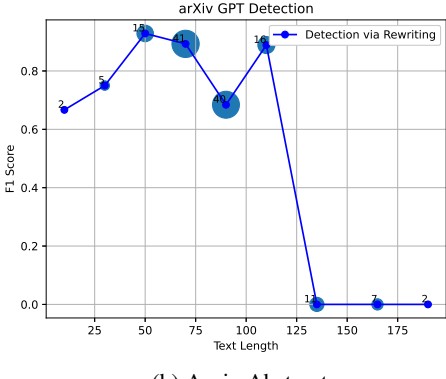

|       (a) Code       |   (b) Arxiv Abstract   |

Figure 9: Detection performance under different length. For both datasets, the performance is high in the beginning, demonstrating the advantage of our approach in tackling sequence that is shorter. However, for longer input, the detection performance drops.

Table 9: Robustness of Our Algorithm to LLM Finetuning. The detection model was only learned on GPT-3.5-Turbo generated data and use GPT-3.5-Turbo for rewriting. We show the results on GPT-3.5-Turbo in the first row. We then directly apply the detector to data generated from GPT-4-Turbo, but use the old, GPT-3.5-Turbo model for rewriting and detection. The detector was never trained on GPT-4-Turbo. Despite a drop in detection effectiveness, our algorithm still outperform the published state-of-the-art zero-shot detector.

| Test Data source | Data Detector | Datasets | | |
|---|---|---|---|---|
| | | Code | Yelp | Arxiv |
| GPT-3.5-Turbo | Ours trained with GPT-3.5-Turbo | 95.38 | 87.75 | 81.94 |
| GPT-4-Turbo | Ours trained with GPT-3.5-Turbo | 83.07 | 79.73 | 74.02 |
| GPT-4-Turbo | Baseline DetectGPT | 70.97 | 66.94 | 66.99 |

Table 10: Robustness on non-native English authors. We show results that train on our Arxiv dataset, and test on the ASAP dataset in the gray row. While the detection score drop a bit from training on ASAP and test on ASAP, we still achieve a F-1 detection score of 81.16, which is only less than 1 point than Arxiv paper. This demonstrate the robustness of our detection algorithm, even trained on the Arxiv papers that are accepted to ICLR, which are high quality written text, our algorithm still generalize well to non-native English writters from grade 8 level.

| Training Source | Testing Source | F-1 Detection Score |
|---|---|---|
| ASAP Dataset | ASAP Dataset | 98.76 |
| Arxiv Dataset | ASAP Dataset | 81.16 |
| Arxiv Dataset | Arxiv Dataset | 81.95 |

Table 11: F1 score for detecting machine-generated paragraphs by combining rewrites from multiple LLMs. We experiment on two datasets.

| Methods | Yelp Reviews | Arxiv Abstract |
|---|---|---|
| GPT-3.5 Only | **87.75** | 81.94 |
| GPT-3.5 + Ada | 85.71 | **92.85** |
| GPT-3.5 + Davinci-003 | 85.53 | 88.40 |
| GPT-3.5 + Davinci-003 + Ada | 81.76 | 90.00 |

Table 12: F1 score for detecting machine-generated paragraphs by edit distance between the rewritten texts from different LLMs as additional features.

| Methods | Yelp Reviews | Arxiv Abstract |
|---|---|---|
| GPT-3.5 Only | **87.75** | 81.94 |
| GPT-3.5 / Ada | 67.85 | 89.21 |
| GPT-3.5 / Davinci | 78.41 | 81.94 |
| Ada / Davinci | 66.25 | **90.51** |

Table 13: Detection performance combining invariance, equivariance, and uncertainty.

| Methods | News | Creative Writing |
|---|---|---|
| Single | **60.29** | **62.88** |
| Combined | 53.72 | 58.18 |

Table 14: Statistical significance of the number of changes (deletions, insertions) done by the selective generative models between humans and machine-generated texts, corresponding to Figure 2 in the main paper. We present the p value by running a one-sided two-sample t tests. The small p-value demonstrates the statistical significance.

| Methods | Invariance | Equivariance | Uncertainty |
|---|---|---|---|
| p-value | 2.19e-13 | 9.21e-7 | 5.47e-16 |

Table 15: Classifier for detecting machine-generated paragraphs. We use the best classifier from logistic regression (LR) and XG Boost for classification.

| | Datasets | | | | | |
|---|---|---|---|---|---|---|
| Methods | News | Creative Writing | Student Essay | Code | Yelp Reviews | Arxiv Abstract |
| Ours Invariance | LR | LR | XGBoost | LR | LR | LR |
| Ours Equivariance | LR | LR | XGBoost | LR | LR | LR |
| Ours Uncertainty | LR | LR | XGBoost | LR | LR | LR |

