# OpenReview forum: "Raidar: geneRative AI Detection viA Rewriting"
_ICLR.cc/2024/Conference — ICLR 2024 poster_

### Official Review · Reviewer_TBk4 · 2023-10-27

**Soundness:** 3 good
**Presentation:** 3 good
**Contribution:** 3 good
**Rating:** 6
**Confidence:** 4

**Summary:**

The paper discusses work on detected AI-generated text using rewriting. The authors formalize their hypothesis which “a generated text from an auto-regressive generative models retains a consistent structure, which another such model will likely to also have a low loss and treat it as high quality”. The proposed method captilizes of this hypothesis through quantifying rewrites in the form of edit distance scoring which the author/s use to train classifiers for the detection phase. The author/s also propose several variations of symbolic modifications including invariance, equivariance, and uncertainty. The author/s select previous well-known detector systems including GPTZero, Ghostbuster, and DetectGPT for the comparison of detection performances on 6 datasets. The author/s report favorable results of the proposed method in in Tables 1 and 2. Other experiments done include robustness in an adversarial setup using paraphrasing, effects of content length, quality of generated data, and type of detection model but these parts seem condensed and not thoroughly discussed.

**Strengths:**

The paper discusses work on an interesting framing of detecting text generated by large language models by gauging how much rewrites (edit distance) it gives to a piece of text with the hypothesis that it considers machine-generated texts, regardless of specific models, as high quality ones compared to human-generated texts. The simplicity of the method is what I appreciate the most with the paper. I also appreciate that the author/s are able to construct a good research plan for the methodology of the study and the formalization of the task. There are some revisions suggested below to improve the structure and organization of the paper, particularly the uniformity and presentation of results.

**Weaknesses:**

The paper starts well with the introduction, statement of motivation, discussion of hypothesis, and proposed method but went downhill with the presentation of results.

First, in the discussion of datasets and model baseline used, the definition and characteristics should definitely be expanded by including the size of the dataset plus other unique characteristics of the data that’s worth mentioning as well as the quantified performance of the baseline models just to give an idea to the readers of their relevance and why they are being used/compared in the study.

Second, the discussion on Section 4 seems to be too condensed and there are sudden insertions of models for comparison. Claude was tested but was not mentioned in the baseline models. Why is that and why is Claude a viable candidate for comparison?

Third, there is no concrete / definitive discussion for the results presented in Table 1 and 2 and these are the bread and butter of the study. The table captions and Section 4.3 simply state the observed increments against the previous systems and nothing more. I would have appreciated it if there are more discussions on particular weaknesses in the texts by previous works which the current work addresses, and give some examples of this. Why are proposed methods get big performance jumps, say in the code dataset and Arxiv dataset, compared to natural language datasets? In the robustness experiments, why are the previous detector models left out? It is important to still consider them in the mix for better comparison of true performance. These discussions should be in order.

The statement “text from the Ada model is most easily detected, possibly due to its discernibly lower quality compared to human-written content” is speculation. Some form of quantitative proof is required for this.

In terms of strengthening the paper, I strongly suggest the author/s to also experiment with second-language or non-native speaker writing corpora such as TOEFL or ESL essay corpus to gauge if the proposed method also exhibits that particular limitation of confusing AI-generated texts with non-native speaker-written texts. This has been a known limitation across all detectors of machine-generated texts. Refer to the existing works (https://arxiv.org/abs/2304.02819) and databases online (https://www.clarin.eu/resource-families/L2-corpora, https://uclouvain.be/en/research-institutes/ilc/cecl/learner-corpora-around-the-world.html) for the data to be experimented with. Regardless of the result, this would be a great addition and insight to the paper.

Lastly, the paper is littered with inconsistencies and wrong format for in-text citations which makes it very confusing to read. For example, it’s better to read referenced text as “Ghostbuster (Verma et al., 2023) instead of Ghostbuster Verman et al (2023)”. It seems like the Verma is included in the name of the model but it’s not. There are also cases where the model is “GPT-3.5-Trubo” and then becomes “GPT-3.5-turbo”. Diagrams in Figure 6 should conform to uniform range of [0.0,1.0] in the y axis.

I still see the importance and contribution of the study but major revision is required for the paper in order for readers to fully comprehend and appreciate the work.

**Questions:**

1. How large is the feature set or dimension then for training the classifiers? Did the author/s conduct and hyperparameter optimization with the classifiers?

2. How is the setup for comparison done for preliminary experiments shown in Table 1 and 2? What classifier was used for these experiments? There were two algorithms mentioned (LR and XGBoost) but it was never specified in the tables. The same way goes for Tables 4 and 5. I’m confused as a reader as to which classifier was used for the presentation of results. Revising this part and enriching the details should be in order.

3. Is the number of changes (deletions, insertions) done by the selective generative models between humans and machine-generated texts statistically significant? The visualization in Figure is nice but this would have been an additional supporting preliminary information to add to the hypothesis.

4. There are six datasets chosen for experimentations but why are only subsets or some of it are used for results presented in Table 3, 4, and 5?

5. The statement in Section 4.4 goes “In Table 5, while all models achieve significant detection performance”. So a statistical test was conducted? Can you give the details for this? Otherwise, I suggest using the word substantial instead.

6. Why is only the Yelp data used for special test cases such as effect of content length? Why not the other datasets as well? This seems very selective without reason.

---

> ### Author Response · Authors · 2023-11-22
> **Thank you for your helpful experiment suggestions in strengthening the paper!**
>
> Thank you for your helpful experiment suggestions in strengthening the paper! We ran them and reported the results below. We also incorporate your revision suggestions. Please let us know if you need any additional information.
>
> **Characteristics of the dataset.**
>
> We show the statistics of the dataset in Table 6,7 in the appendix.
>
> **Reason for choosing the studied datasets.**
>
> Our dataset selection was driven by the need to address emerging challenges and gaps in current research. We incorporated news, creative writing, and essays from the established Ghostbuster paper to maintain continuity with prior work. Recognizing the growing capabilities of Language Learning Models (LLMs) like ChatGPT in generating code, and the accompanying security issues, we included code as a novel and pertinent text data type. Additionally, we analyzed Yelp reviews to explore LLMs' potential for generating fake reviews, a concern overlooked in previous studies, which could significantly influence public opinion about businesses. Lastly, we included arXiv data to address recent concerns over the use of GPT in academic writing, reflecting on its ethical implications.
>
> **Justification for Additional LLM generated text, such as from Claude, included for comparison.**
>
> To evaluate the robustness of our model, we expanded our testing to include LLMs from companies beyond OpenAI. We incorporated Claude data from the Ghostbuster paper, as suggested by reviewer 3, because Claude, developed by Anthropic, differs from OpenAI's models, offering a diverse robustness check. However, due to Ghostbuster's data limitations, we could only obtain one dataset for Claude. Furthermore, we integrated LLaMA, an LLM from Meta, and conducted extensive tests across six datasets. The results demonstrate our method's ability to effectively identify content generated by LLaMA, despite not being trained on it. This highlights our approach's capability to reliably detect machine-generated content across various company products.
>
>
> | LLM Model Used for Text Generation | In Distribution Code | In Distribution Yelp | In Distribution arXiv | Ours Out of Distribution Code | Ours Out of Distribution Yelp | Ours Out of Distribution arXiv | DetectGPT Code | DetectGPT Yelp | DetectGPT arXiv |
> |------------------------------------|----------------------|----------------------|-----------------------|-------------------------------|--------------------------------|---------------------------------|----------------|----------------|-----------------|
> | GPT-3.5-turbo                      | 95.38                | 87.75                | 81.94                 | 81.07                         | 71.43                          | 48.74                           | 70.97          | 65.45          | 66.67           |
> | LLAMa 2                            | 98.46                | 89.31                | 97.87                 | 70.96                         | 89.30                          | 74.41                           | 68.42          | 67.24          | 66.67           |
>
> **Discussion in Table 1,2**
>
> Thank you for your feedback. We will update in our revision.
> Our investigation into the significant performance improvement on code and arXiv datasets revealed that these are highly specialized tasks. LLMs like GPT often fall short in matching the quality of human-generated content in these areas, leading to a more pronounced disparity between machine-generated and human-generated texts, especially in rewriting scenarios. Previous methods, which primarily used smaller models such as OPT and GPT-2 as their score functions, struggled more with complex tasks like coding and academic writing. In contrast, our method utilizes the advanced capabilities of GPT-3.5, which better handles these challenging tasks, resulting in superior outcomes than prior approach. Additionally, for a comprehensive robustness evaluation, we included the state-of-the-art DetectGPT in Table 4, allowing for a direct comparison of its effectiveness.
>
> **Speculation Sentence**
>
> We removed the speculation from the revision.

---

> ### Author Response · Authors · 2023-11-22
> **Continue**
>
> **Experiment with second-language and non-native speaker**
>
> Thank you for your suggestion in strengthening our paper. We have cited the paper [1] and conducted the experiment in our paper. We utilized 200 examples from the ASAP dataset, featuring TOEFL essays authored by non-native English speakers. Our experiments, detailed in the subsequent table, involved training on the ASAP dataset and testing on it, as well as training on our original arXiv dataset and testing on ASAP in an out-of-domain context. Our method, not relying on word semantics, showed strong generalization to the previously unseen ASAP data. Our approach won't mistakenly identify non-native speaker-written texts as AI-generated, given that LLMs might also classify these as lower quality, prompting substantial rewrite edits. Our empirically results below validate this.
>
> | Training Source | Testing Source | F-1 Detection Score |
> |-----------------|----------------|---------------------|
> | ASAP Dataset    | ASAP Dataset   | 98.76               |
> | Arxiv Dataset   | ASAP Dataset   | 81.16               |
> | Arxiv Dataset   | Arxiv Dataset  | 81.95               |
>
> **Q1: Feature size and hyperparameter for training classifier.**
>
> The Feature size is 28 and we did model selection for the classifier.
>
> For each input, our method generates one Levenshtein score and three sets of bag-of-words edit features—derived from one, two, and three bags of words—resulting in four distinct metrics per prompt. We apply seven different rewriting prompts to each input, leading to a total of 28 features. To process these features, we employed the Logistic Regression and XGBoost algorithms using their default configurations in the Sklearn package. After evaluating their performance, we selected and presented the results from the classifier that demonstrated better efficacy. Based on our results, we use logistic regression for all our experiments except for the essay dataset.
>
> **Q2: Classifier Setup for Table 1,2,3,4.**
>
> We use logistic regresssion for all experiments except for Table 1's student essay, which we used XGBoost. We run both and use the one that performs higher. We find XGboost performs better on student essay tasks, so we used XGBoost for this task in Table 1 and logistic regression for all other experiments.
>
> **Q3: Number of change is significant.**
>
> To access the significance of the number of the change, we calculate the one-sided two-sample t tests and calculate the p-value for the changes between human-generated text and machine-generated text (corresponding to the histogram in Figure 2). The small p-value demonstrate that the change is very significant.
>
> | Methods  | Invariance | Equivariance | Uncertainty |
> |----------|------------|--------------|-------------|
> | P-Value  | 2.19e-13   | 9.21e-7      | 5.47e-16    |
>
>
> **Q4: Dataset choice in Table 3-5.**
>
> Thank you for your question. In our experiments for Tables 3 and 4, adjustments in the generated texts were necessary, involving changes to both the prompts and the generation models. However, since the datasets comprising News, Creative Writing, and Student Essays, as released by Ghostbuster, are fixed and do not permit such manipulations, we were unable to conduct analyses on these datasets for these tables.
>
> Following your suggestion, for Table 5, we have provided comprehensive results since we can vary the rewriting model, ensuring that the entire scope of our findings and analyses for this specific table is thoroughly detailed and accessible. We show the complete results below:
>
> | LLM for Rewriting | News  | Creative Writing | Student Essay | Code  | Yelp  | Arxiv |
> |-------------------|-------|------------------|---------------|-------|-------|-------|
> | Ada               | 55.73 | 62.50            | 57.02         | 77.42 | 73.33 | 71.75 |
> | Text-Davinci-002  | 55.47 | 60.59            | 58.96         | 82.19 | 75.15 | 59.25 |
> | GPT-3.5 turbo     | 60.29 | 62.88            | 64.81         | 95.38 | 87.75 | 81.94 |
>
>
>  **Q5: Replace word ”significant” with “substantial”**
>
> Thank you. We updated.
>
> **Q6: Additional results for effect of content length**
>
> Thank you for your suggestion. We've included supplementary results on the impact of content length in Appendix Figures 7, 8, and 9. Generally, longer content tends to enhance detection performance, with the notable exceptions of code and arXiv papers. We conjecture that this deviation may be attributed to LLMs struggling with rewriting longer pieces of code or academic papers, likely due to the intrinsic complexity associated with these specific tasks.

---

### Official Review · Reviewer_cUTp · 2023-10-28

**Soundness:** 2 fair
**Presentation:** 3 good
**Contribution:** 3 good
**Rating:** 6
**Confidence:** 5

**Summary:**

This paper provides a new approach for detecting LLM-generated text, based on the insight that LLMs will make minimal edits when asked to rewrite their own generations. The paper proposes three related detection models called Invariance, Equivariance, and Uncertainty and shows that they outperform state-of-the-art detectors such as Ghostbuster, DetectGPT, and GPTZero on a variety of paragraph-level detection tasks. The paper also introduces three new domains of generated code, Yelp reviews, and arXiv abstracts, finding that the model again outperforms existing baselines. While the paper has some missing details and would benefit from additional rounds of proofreading, the core idea and results lead me to recommend acceptance.

**Strengths:**

- The core idea in this work, that LLMs are less likely to make extensive edits to LLM-written text than human-text, is an intuitive and seemingly very reasonable approach.
- The paper obtains impressive performance compared to state-of-the-art models (Ghostbuster and GPTZero). In particular, the model seems to work especially well on short documents, which is a known failure mode for existing approaches
- The paper introduces three new detection datasets (Yelp reviews, code, arXiv abstracts) and evaluates on them in addition to an existing

**Weaknesses:**

1. The paper is missing some important details. For example, the paper does not provide a complete explanation of how the human reviews from the Yelp dataset were selected, nor does it provide a full explanation of how GPT 3.5 was prompted to generate data in the code/Yelp/arXiv domains. The paper also does not describe which scoring model was used for the DetectGPT results.
2. I believe the results on OOD generalization are slightly misleading. Because DetectGPT and GPTZero should produce the same scores in-domain and out-of-domain, GPTZero actually has the highest score for the OOD creative writing task
3. Table 4 (robustness across generation models) is difficult to interpret without results from the baselines. Although the model seems to generalize well to generators it was not originally trained on (ada and davinci), these are also weaker models which (as stated in the table caption) results in an easier detection problem. Additionally, generalization results in the paper would benefit from the inclusion of models which are not from OpenAI, since these are likely more similar to one another than models like Claude, LLaMA, etc.

MISCELLANEOUS SUGGESTIONS
- A small suggestion on related work: the introduction starts that token log probabilities are not available in black-box models such as GPT-3.5 and GPT-4, which is true. I think the paper would benefit from distinguishing “scoring models” from “target models” when discussing how this affects related work. DetectGPT is designed to only work when the scoring and target model are the same (cf. Figure 6 in their paper), although Ghostbuster operates under the assumption that the scoring and target model are different (but still requires access to generated documents from the target model).
- Minor: Figure 6 text is too small to read and would benefit from error bars
- The paper has a large number of typos and grammatical errors. Although this does not affect my review score, the paper would benefit from an additional round of proofreading.

**Questions:**

1. Are all detection tasks in this paper at the paragraph level? How does the median length of documents vary across tasks? Did you try the method for any longer documents?
2. Did you experiment with combining the invariance, equivariance, and uncertainty features?
3. How was the impact of context length experiment run? In Figure 5, how many examples were included for each point on the plot?

---

> ### Author Response · Authors · 2023-11-22
> **Thank you for your thoughtful review and suggestions on the experiments**
>
> Thank you for your thoughtful review, questions, and suggestions on the experiments. We ran them and presented them below.
>
> **1. How Human Reviews from the Yelp was selected**
>
> We select the first 2000 reviews from the Yelp Review dataset.
>
> **1. How GPT 3.5 was prompted to generated data**
>
> **Code:** Human eval dataset offers code specification and the completed code for each data point. We first use GPT to generate a detailed description of the function of the code by prompting it with “Describe what this code does {code specification}{code}”. The result, termed {pseudo code}, is an interpretation of the code. Subsequently, we prompt GPT with "I want to do this {pseudo code}, help me write code starting with this {code specification}," to generate Python code that adheres to the given input-output format and specifications. This way, we create the AI-generated code data.
>
> **Yelp:** When tasked with crafting a synthetic Yelp review, prompting GPT-3.5 with "Help me write a review based on this {original review}" resulted in verbose and lengthy text. However, we discovered that using the prompt "Write a very short and concise review based on this: {original review}" yielded the most effective and succinct AI-generated reviews.
>
>
> **Arxiv:** In our experiment with Arxiv data, which includes titles and abstracts, we synthesized abstracts by using the title and the first 15 words of the original abstract. We employed the prompt “The title is {title}, start with {first 15 words}, write a short concise abstract based on this:”, which successfully generated realistic abstracts that align with the titles.
>
> **1. Which scoring model was used for DetectGPT:**
>
> We used the facebook/opt-2.7B model, which boasts a larger parameter count compared to the GPT-2 model with its 1.5B parameters.
> We use the T5-3B model for the mask filling model, which is stated to be used for almost all experiments in the original paper.
>
> **2. Discussion on Results for OOD Generalization**
>
> Thank you for bringing this out. Yes, DetectGPT and GPTZero display consistent performance in both in-domain and out-of-domain tests, with our model outperforming them in out-of-domain scenarios, except for GPTZero in Creative Writing. However, The reliability of GPTZero's zero-shot capabilities for Creative Writing is uncertain due to its blackbox nature and unknown training data. Table 1 shows GPTZero's performance variability across tasks, notably dropping to 25 in Yelp Reviews, suggesting weak out-of-domain generalization and large variation over tasks.
> We will include those discussions over this in our revision.
>
>
> **3. Baselines for Table 4 Results**
>
> Following your suggestion, we add DetectGPT baseline and updated Table 4:
>
> | LLM Model Used for Text Generation | In Distribution Code | In Distribution Yelp | In Distribution arXiv | Ours Out of Distribution Code | Ours Out of Distribution Yelp | Ours Out of Distribution arXiv | DetectGPT Code | DetectGPT Yelp | DetectGPT arXiv |
> |------------------------------------|----------------------|----------------------|-----------------------|-------------------------------|--------------------------------|---------------------------------|----------------|----------------|-----------------|
> | Ada                                | 96.88                | 96.15                | 97.10                 | 62.06                         | 72.72                          | 70.00                           | 67.39          | 70.59          | 69.74           |
> | Text-Davinci-002                   | 84.85                | 65.80                | 76.51                 | 75.41                         | 51.06                          | 60.00                           | 66.82          | 68.70          | 66.67           |
> | GPT-3.5-turbo                      | 95.38                | 87.75                | 81.94                 | 81.07                         | 71.43                          | 48.74                           | 70.97          | 65.45          | 66.67           |
> | GPT-4-turbo                        | 80.00                | 83.42                | 84.21                 | 83.03                         | 79.73                          | 74.02                           | 70.97          | 69.64          | 66.99           |
> | LLAMa 2                            | 98.46                | 89.31                | 97.87                 | 70.96                         | 89.30                          | 74.41                           | 68.42          | 67.24          | 66.67           |

---

> ### Author Response · Authors · 2023-11-22
> **Continue**
>
> **3. Generalization results to text generated from stronger LLM**
>
> Thank you for your suggestion. We added experiment with data generated from GPT-4, which is a stronger model than GPt-3.5.
>
> | LLM Model Used for Text Generation | In Distribution Code | In Distribution Yelp | In Distribution arXiv | Ours Out of Distribution Code | Ours Out of Distribution Yelp | Ours Out of Distribution arXiv | DetectGPT Code | DetectGPT Yelp | DetectGPT arXiv |
> |------------------------------------|----------------------|----------------------|-----------------------|-------------------------------|--------------------------------|---------------------------------|----------------|----------------|-----------------|
> | GPT-3.5-turbo                      | 95.38                | 87.75                | 81.94                 | 81.07                         | 71.43                          | 48.74                           | 70.97          | 65.45          | 66.67           |
> | GPT-4-turbo                        | 80.00                | 83.42                | 84.21                 | 83.03                         | 79.73                          | 74.02                           | 70.97          | 69.64          | 66.99           |
>
> **3. Results generalize to models not from OpenAI, such as LLaMa and Claude**
>
> In the paper, we have included text generated from Claude that is provided by Ghostbuster paper, that we show F1 score of 57.80. In addition, here we show results from LLaMA v2-70B that is trained by Meta:
>
> | LLM Model Used for Text Generation | In Distribution Code | In Distribution Yelp | In Distribution arXiv | Ours Out of Distribution Code | Ours Out of Distribution Yelp | Ours Out of Distribution arXiv | DetectGPT Code | DetectGPT Yelp | DetectGPT arXiv |
> |------------------------------------|----------------------|----------------------|-----------------------|-------------------------------|--------------------------------|---------------------------------|----------------|----------------|-----------------|
> | GPT-3.5-turbo                      | 95.38                | 87.75                | 81.94                 | 81.07                         | 71.43                          | 48.74                           | 70.97          | 65.45          | 66.67           |
> | LLAMa 2                            | 98.46                | 89.31                | 97.87                 | 70.96                         | 89.30                          | 74.41                           | 68.42          | 67.24          | 66.67           |
>
>
> **Suggestion 1: Related work and writing:**
>
> We will update as suggested.
>
> **Q1: Detection tasks at paragraph level?**
>
> Yes, our detection tasks are at the paragraph level. We calculate the median length of documents across tasks and shown in Table 6,7, and also below:
> |                      | News | Creative Writing | Student Essay | Code | Yelp | Arxiv |
> |----------------------|------|------------------|---------------|------|------|-------|
> | Dataset Size         | 730  | 973              | 22172         | 164  | 2000 | 350   |
> | Median Length        | 38   | 21               | 96            | 96   | 21   | 102   |
> | Minimum Length       | 2    | 2                | 16            | 2    | 6    | 19    |
> | Maximum Length       | 122  | 295              | 1186          | 78   | 1006 | 274   |
>
> **Q2: Experiment combining invariance, equivariance, and uncertainty**
>
> Yes, following your suggestion, we tried combining them by concatenating the features over two datasets. Yet we did not achieve better results by combining.
>
> | Methods   | News  | Creative Writing |
> |-----------|-------|------------------|
> | Single    | 60.29 | 62.88            |
> | Combined  | 53.72 | 58.18            |
>
> **Q3: Experimental design and number of examples for each point in Fig 5.**
>
> Thank you for your question. We updated the Figure 5 to show the number of samples for each point via the size of the dot. We run the detection over Yelp test set which contains 800 samples. The number of points for each point in Figure 5 is [51, 204, 164, 135, 82, 84, 80].

---

### Official Review · Reviewer_LoWc · 2023-11-01

**Soundness:** 3 good
**Presentation:** 4 excellent
**Contribution:** 4 excellent
**Rating:** 8
**Confidence:** 3

**Summary:**

The paper proposes to use rewriting as a task to detect whether a text is generated or not. They show that LLMs rewrite the text less when given a LLM-written text as opposed to a human-written text. The paper introduces a method that is based on prompting LLMs to rewrite text and calculating the editing distance of the output. This approach significantly improves the detection of LLM-written texts across domains.

**Strengths:**

* The paper tackles on a task that is very important to help mitigate the misuse of LLMs.

* The method proposed is simple yet effective. It is also very intuitive since it is actually also human nature to prefer (=critic less) your own writing than others.

* The paper is very well written.

**Weaknesses:**

* It would be nice to actually check the quality of human/LM written ext., i.e. which between the human and LM text do another set of humans prefer? It could be that the LM-written text is actually better quality (and thus requires fewer rewrites).

* It would be nice to include a discussion on how this method fares when the LLM is instruction-tuned to do rewrites the way humans do rewrites. Given that text rewriting is a widely used use case for LLMs, it would not be a surprise if these models are instruction-tuned on a task mixture that includes text rewriting.

**Questions:**

See weaknesses above.

---

> ### Author Response · Authors · 2023-11-22
> **Thank you for your thoughtful review and useful suggestions.**
>
> We appreciate your thoughtful review and we hope we addressed your concerns. Please let us know if you'd like any further information.
>
> **1. Quality of human/LM written text**
>
> Thank you for your suggestion. Yes, we manually checked the quality of human/LM text. In general, machine generated text is similar or less preferred by human than human-written text, suggesting that LLM perceive text quality in a different way than human. Due to the time limit, we conducted a user study on Arxiv and Yelp datasets over 40 random examples over 3 users due to the time constraint, produced by GPT-3.5. We show the results below. We will conduct experiments over more users.
>
> | | Yelp Reviews | Arxiv Abstract |
> |---------------------------------------------------------------|--------------|----------------|
> |                % that Machine Generated Text are Preferred Human Written Text                                                | 53.3%        | 26.7%          |
>
> **2. Discussion on LLM is finetuned to mimic human rewrite**
>
> Thank you for your suggestion. In Section 4.4 of our paper, we explored a related concept by prompting the model to emulate human-style writing. Our findings indicated a reduction in detection effectiveness under these conditions. This scenario can be viewed as a rudimentary form of LLM fine-tuning, where the prompt is adjusted rather than the model's parameters. Extrapolating from this trend, it's plausible that more extensive LLM fine-tuning would further diminish the efficacy of our detection method. Consequently, future research should focus on adapting detection models to effectively identify content from fine-tuned LLMs that closely mimic human rewrites. As our study is pioneering in examining rewriting for detection, and considering the high costs associated with fine-tuning LLMs, we suggest this as future research.

---

### Official Review · Reviewer_WaeQ · 2023-11-09

**Soundness:** 3 good
**Presentation:** 4 excellent
**Contribution:** 3 good
**Rating:** 6
**Confidence:** 4

**Summary:**

This work addresses the problem of detecting AI-generated text. It makes a hypothesis that LLMs, when asked to rewrite, are more likely to modify human-generated text than AI-generated text. The hypothesis is based on the observation that AI-generated text is generally of higher quality than the average human-generated text and therefore needs fewer modifications. The work proposes a method for detecting AI-generated text based on the above hypothesis. The proposed method essentially involves asking LLM to rewrite the text and computing the edit distance between the input and the rewritten text. In this context the work introduces the notions of invariance, equivariance and uncertainty. Invariance is measured as the modification distance between the input and the LLM-rewritten input. Equivariance is measured as the modification distance between the rewritten text and the text obtained by transforming the input, rewriting and finally undoing the transformation. Uncertainty is measured as the average modification distance between different rewritings of the input by the same LLM with the same prompt.

The work reports results from an experimental study of the proposed approach on several datasets and compares with several baselines. Invariance based scoring gives significantly better results than the baselines while equivariance and uncertainty based scoring also do well. The work also reports results from a study on robustness, source of generated data, different LLMs for rewriting, and impact of prompts used for rewriting, and the length of the input text.

**Strengths:**

1. Simple method for detecting AI-generated text that essentially consists of asking LM to rewrite the input and then computing edit-distance between the input and the rewritten text. Computationally simple.

2. Significant improvement in detection effectiveness over the baselines.

3. Interesting study on robustness, source of generated data, different LLMs for rewriting, impact of prompts used for rewriting, and the length of the input text.

**Weaknesses:**

1. Proposed method is critically dependent on LLM. Any change to LLM due to continual fine-tuning with new data might have unknown consequences for the detection method. The method might not be robust to LLM fine-tuning.

2. Though the proposed method is simple and computationally low cost, there is a cost associated with every call to the detection algorithm because of the calls made to LLM for rewriting. This might not be acceptable in some scenarios it is cheaper and desirable to have a model decoupled for LLM after training.

**Questions:**

1. In Table 1 you write "The results are in domain testing, where the model has been trained on the same domain." but you also use the term "in distribution" Are these equivalent terms?  I assume you consider each of the datasets as a different domain (e.g. news domain vs creating writing).  What is the difference between in-distribution and in-domain?

2. In Table 2, what is the domain for training and what is the domain for testing?

3. Would the effectiveness of detection improve further if multiple LLMs are asked to rewrite the input text and invariance scores for each LLM are used as features for training the logistic regression classifier? Also, would edit distance between the rewritten texts from different LLMs be useful as additional features for such a classifier?

---

> ### Comment · Reviewer_WaeQ · 2023-11-22
> **Author Response**
>
> The authors have not responded to the reviews, at least till now. My scores will therefore remain unchanged.

---

> > ### Author Response · Authors · 2023-11-22
> > **Sorry for the delay**
> >
> > We are sorry for the delay. We have spent a lot of time in running the experiments suggested by all the reviewers. We ran them and incorporated them to strengthen the paper.

---

> ### Author Response · Authors · 2023-11-22
> **Thank you for your thoughtful review and constructive suggestions.**
>
> We thank the reviewer for their thoughtful review. We are glad that the review found our approach simple and effective, with interesting studies. We address the reviewer’s concern below:
>
> **1. Method Robustness to different LLM, such as finetuning**
>
> Thank you for your question. We ran the experiment on GPT-3.5 and GPT-4. GPT-4 can be treated as an advanced, continual fine-tuned LLM with new real-world data. We show the results In Table 7 in the appendix, as well as follows.
>
> Our detector, trained only on GPT-3.5, can still work well on detecting GPT-4-Turbo generated data. Our method outperforms  established state-of-the-art DetectGPT. This means our method is robust to LLM finetuning.
>
> | Test Data source | Data Detector | Code | Yelp | Arxiv |
> |------------------|---------------|------|------|-------|
> | GPT-3.5-Turbo    | Ours trained with GPT-3.5-Turbo | 95.38 | 87.75 | 81.94 |
> | GPT-4-Turbo    | Ours trained with GPT-3.5-Turbo | 83.07 | 79.73 | 74.02 |
> | GPT-4-Turbo      | Baseline DetectGPT              | 70.97 | 66.94 | 66.99 |
>
> In Table 4 in our paper, we also evaluate our method over detecting text generated from 5 different LLMs, demonstrating the robustness of our detector to different LLMs.
>
> **2. The cost associated with calling the LLM API**
>
> Thank you for bringing up the cost. While there is no free algorithm, in Table 5, we have shown using cheaper OpenAI models, such as Ada which is 10 times cheaper for detection. Given the price reduction of OpenAI’s recent API cost, our approach will be more cost efficient.
>
> In addition, to avoid the cost of OpenAI API, we also show that open-source LLaMA-7B, which can be run on a single RTX 3090 Ti GPU, can also be integrated into our framework, avoiding the fee to pay for OpenAI API calls. As shown below, it only leads to a performance drop of a few points, still achieving up to 85 points F-1 detection accuracy score.
>
> | LLM for Rewriting | News  | Creative Writing | Student Essay | Code  | Yelp  | Arxiv |
> |-------------------|-------|------------------|---------------|-------|-------|-------|
> | Ada               | 55.73 | 62.50            | 57.02         | 77.42 | 73.33 | 71.75 |
> | Text-Davinci-002  | 55.47 | 60.59            | 58.96         | 82.19 | 75.15 | 59.25 |
> | GPT-3.5 turbo     | 60.29 | 62.88            | 64.81         | 95.38 | 87.75 | 81.94 |
> | LLAMA 2           | 56.26 | 61.88            |       60.48      | 85.33 | 74.85 | 72.59 |
>
> **Q1: Term Usage, the difference between in-distribution and in-domain**
>
> Sorry for the confusion. Yes, they are the same terms. We will use a consistent “in domain” word in our revision.
>
> **Q2: The training and testing domain for Table 2**
>
> For all experiments in Table 2, we use logistic regression, and use the same source and target for invariance, equivariance, and uncertainty.
> News: train on Creative Writing and test on News.
> Creative Writing: train on News and test on Creative Writing.
> Student Essay: train on News, and test on student Essay.
>
> **Q3: The effectiveness of using multiple LLM to rewrite the input text**
>
> Thank you for your suggestion. We conducted this study on Yelp and Arxiv, results show that the suggested method can improve the detection score by up to 10 points. We show the results in Table 9 and also follows:
>
> | Methods               | Yelp Reviews | Arxiv Abstract |
> |-----------------------|--------------|----------------|
> | GPT-3.5 Only          | 87.75        | 81.94          |
> | GPT-3.5 + Ada         | 85.71        | 92.85          |
> | GPT-3.5 + Davinci-003 | 85.53        | 88.40          |
> | GPT-3.5 + Davinci-003 + Ada | 81.76  | 90.00          |
>
>
> **Q3: Edit distance between the rewritten texts from different LLMs as additional features.**
>
> Yes, they can be useful features. Following your suggestion we run the experiment on Yelp and arxiv. We can achieve up to to 8 points gain on arxiv dataset. We also find the improvement is dataset dependent.
>
> | Methods               | Yelp Reviews | Arxiv Abstract |
> |-----------------------|--------------|----------------|
> | GPT-3.5 Only          | 87.75        | 81.94          |
> | GPT-3.5 / Ada         | 67.85        | 89.21         |
> | GPT-3.5 / Davinci-003 | 78.41       | 81.94        |
> | Ada / Davinci-003 + Ada | 66.25  | 90.51          |

---

### Author Response · Authors · 2023-11-23
**We ran the experiments asked by the reviewer and updated the paper. Edits are in blue.**

We thank all reviewers for their valuable feedback. We are glad that all reviewers find our method simple and effective. We ran all the experiments asked by the reviewers and integrated them to strengthen our paper. We indicate our paper edits in blue.

---

### Meta-Review · Area_Chair_JJE2 · 2023-12-07

**Metareview:**

The paper introduces a simple yet effective method for detecting AI-generated text by leveraging language models (LLMs) to rewrite input text and then computing the edit distance between the original and rewritten texts. The method shows significant improvement in detection effectiveness compared to existing baselines and demonstrates robustness across different data sources and LLMs. However, reviewers have raised concerns about the method's dependency on LLMs, potential cost implications due to LLM calls, and gaps in the paper's presentation and experimental design. The authors have responded to these concerns, leading to some improvement in the reviewers' ratings.

**Justification For Why Not Higher Score:**

Changes in LLMs due to continual fine-tuning could unpredictably affect the detection accuracy, making the method less robust over time. Also authors should better highlight limitations of their approach for example against adversarial paraphrasing.
Additionally, the cost of LLM calls for rewriting, although the method itself is computationally simple, might be prohibitive in certain scenarios. The paper also lacks comprehensive analysis in areas like the quality comparison between human and LLM-written text, the impact of instruction-tuning on LLMs, and the exploration of more diverse datasets, including non-native speaker corpora.

**Justification For Why Not Lower Score:**

The paper has an innovative approach to AI-generated text detection and it demonstrates substantial improvements  over existing methods.

---

### Decision · Program_Chairs · 2024-01-16

Accept (poster)